# Modulation of Cell–Cell Interactions in Drosophila Oocyte Development

**DOI:** 10.3390/cells9020274

**Published:** 2020-01-22

**Authors:** Matthew Antel, Mayu Inaba

**Affiliations:** Department of Cell Biology, University of Connecticut Health Center, Farmington, CT 06030, USA; antel@uchc.edu

**Keywords:** *Drosophila*, oocyte development, cell-cell interaction, local signaling

## Abstract

The *Drosophila* ovary offers a suitable model system to study the mechanisms that orchestrate diverse cellular processes. Oogenesis starts from asymmetric stem cell division, proper differentiation and the production of fully patterned oocytes equipped with all the maternal information required for embryogenesis. Spatial and temporal regulation of cell-cell interaction is particularly important to fulfill accurate biological outcomes at each step of oocyte development. Progress has been made in understanding diverse cell physiological regulation of signaling. Here we review the roles of specialized cellular machinery in cell-cell communication in different stages of oogenesis.

## 1. Introduction

Precise cell-cell interactions are fundamental to animal development and maintenance of functional tissues; yet studying such cell interactions in in vivo settings proves challenging. How do cells identify their target to communicate in a complex tissue? How is signal range determined? Specialized cellular mechanisms, such as usage of membrane tubular structures, membrane reorganization, and modification of extracellular environment, have been identified to modulate various signaling pathways. *Drosophila* oogenesis provides an outstanding model system to understand the spatial-temporal regulation of signaling based on its well-defined anatomy and small organ size. Recent studies with advanced imaging techniques have started to unveil different mechanisms by which different cell types employ distinct modes of signaling by modifying their cellular machinery (Table 1).

A *Drosophila* ovary consists of 16–20 ovarioles [1]. Each ovariole contains follicles or egg chambers, which are ordered developmentally from anterior to posterior. There are 14 stages of egg chamber development in oogenesis (Figure 1A). At the anterior end of the ovariole, the germarium houses stem cells and their earliest descendants of both the germline and supporting somatic cell lines (Figure 1B) [1,2]. The germarium is divided into four regions (1, 2a, 2b, and 3), based on morphological differences (Figure 1B). In region1, located at the germarium tip, 2–3 germline stem cells (GSCs) contact a cluster of somatic cap cells. A GSC divides asymmetrically to self-renew and give rise to a cystoblast. The cystoblast undergoes four rounds of incomplete divisions to generate an interconnected 16-cell germline “cyst” within the germarium [1,2,3]. One of these cells becomes an oocyte in region 2b and localizes to the posterior of the cyst. The remaining 15 cells become nurse cells (Figure 1B,C) [1,2,3].

Between regions 2a and 2b of the germarium, the cyst encounters somatic follicle stem cells (FSCs). FSCs divide to form a monolayer of follicle cells which encapsulates the cyst. After the cyst exits from germarium, the egg chamber emerges as a stage 2 egg chamber. Follicle cells are specified into several different cell types by receiving positional cues from the encapsulated germline cyst [3,4]. Each follicle cell type also signals back to the germline cyst to support the oocyte growth, maturation, and polarization (Figure 1C–H) [5].

Well-conserved signaling pathways, such as the Bone morphogenetic protein (BMP), Notch/Delta, Hedgehog (Hh), Wnt and Janus kinase/signal transducer and activator of transcription (JAK/STAT), are utilized repeatedly throughout the oogenesis (Table 1). In other words, different combinations of the cells within the same tissue sometimes use the same signaling pathway, indicating that spatiotemporal modification of signaling is critical to prevent communication between the wrong partners (Figure 2). Below, we summarize the cell-cell interactions during *Drosophila* oogenesis with an emphasis on the cellular structure and/or component that may enables/modifies the cell-cell communications.

## 2. Niche-Stem Cell Interaction Utilizes Specialized Cellular Protrusions

The stem cell niche is a specialized environment that promotes stem cell maintenance. Niche signals are thought to be “short-range” in nature, as these molecules influence the stem cell, but not nearby differentiating daughter cells. Specialized cellular protrusions have been reported to play essential roles in this process [22].

In adult females, oogenesis continues to occur throughout the female’s life relying on the activity of GSCs. Asymmetric division of a GSC is crucial to sustain oogenesis, while maintaining the GSC population. Cap cells secrete Decapentaplegic (Dpp), a Bone Morphogenic Protein (BMP) pathway ligand, which promotes GSC self-renewal. Only GSCs adjacent to cap cells receive high levels of Dpp to activate the downstream pathway (Figure 1B). Cystoblasts, displaced one cell diameter away from the niche, do not receive, or receive much less Dpp. Thus, the establishment and maintenance of a sharp Dpp gradient across the anterior part of the germarium is critical to limit the niche space. Multiple mechanisms have been reported to restrict and ensure Dpp signal range within one cell diameter wide so that it only activates GSCs but not cystoblasts [6,23,24,25,26,27,28,29,30,31,32,33,34].

A recent work demonstrated that Dpp is tethered by heparan sulfate proteoglycan (HSPG), Dally at the anterior face of the cap cell cluster, several micrometers away from the GSC-cap cell interface (Figure 1E) [6]. As there is a gap from the GSC-cap cell interface to Dpp reservoir, the GSC utilizes cellular protrusions, the actin-based and microtubule-rich projections (the latter is termed “cytocensors”, Figure 1E) [6]. Interestingly, these two types of protrusions share distinct roles in BMP signaling. While the actin-based protrusions promote Dpp signaling, the microtubule-rich cytocensors are reported to attenuate BMP signaling. 

Microtubule-rich protrusions were also reported in *Drosophila* testicular GSCs, termed MT (microtubule-based)-nanotubes [35]. These stem cell protrusions may provide the solution for highly specific niche-stem cell signaling. It is still poorly understood how these protrusions are formed and how trafficking in and out of these protrusions is regulated. Moreover, whether other stem cell populations use similar protrusions remains an open question. 

## 3. Escort Cell and Germline Interaction

Escort cells are the somatic cell population located at the basement membrane of the anterior half of the germarium, closely associating with each GSC, cystoblast and developing germline cyst (Figure 1B) [36]. Anterior-most escort cells couple with GSCs to maintain stemness, while posterior escort cells encapsulate germ cells to promote differentiation. Several studies have demonstrated actin-dependent mechanisms in escort cell and germline interaction. The actin cortex underlying the plasma membrane plays a key role in plasma membrane reorganization to define cell shape. 

Apically located escort cells are in direct contact with cap cells and receive Hh signals from cap cells. This signal is in turn required for the anterior-most escort cell’s ability to maintain GSC cell fate [7]. Cytonemes are actin-rich, thin membrane protrusions originally found in *Drosophila* imaginal discs [37]. In the germarium, cap cell utilizes similar structure for Hh ligand delivery toward escort cells (Figure 1F) [7]. Hh delivering cytonemes extend until they successfully engage the signal [7]. Since escort cells are located close to cap cells, these cytonemes do not mediate distant signaling between cells. Instead, cytonemes may penetrate target cells to create more surface area for receptor/ligand interaction. Alternatively, when an escort cell does not respond to Hh delivery, extended cytonemes can find different escort cells. Such mechanisms may enable a cell to flexibly respond to shortage of ligand or insufficient target cell response. Therefore, the cytoneme-mediated signaling may act to buffer alteration of extracellular conditions to secure the size of the GSC population. 

Posterior escort cells extend their membranes between tightly packed germ cells, and alternate between encapsulating and releasing them. In the absence of these extensions, germline differentiation fails [38,39]. Germ cells secrete EGF ligand Spitz to activate EGFR signaling in escort cells, which in turn supports the escort cell extensions [40]. JAK/STAT and EGFR signaling sustain escort cell protrusiveness and flexibility by regulating the activity of different small Rho-GTPases (Cdc42 and Rho1), the actin remodeling factors [10]. This study explained well how multiple signaling pathways coordinate dynamic cell shape changes [10]. A subset of protrusions are dynamic, continuously retracting and extending, whereas others become stabilized. This allows escort cells to maintain extensive contacts with germ cells while also permitting the movement of differentiating GSC descendants in a posterior direction [10], explaining how escort cell membranes pass germ cells from one escort cell to the next as previously demonstrated using long-term live imaging [36]. Moreover, such posterior allocation ensures the displacement of differentiating germ cells away from the anterior maintenance niche which may help to prevent unfavorable dedifferentiation of these cells. 

The actin cytoskeleton regulates receptor signaling on the plasma membrane via modulating receptor mobility, crosstalk and signaling pathway activation [41]. Whether such dynamic membrane reorganization also regulates receptor-mediated signaling between germline and soma remains unknown. In addition, signal(s) from escort cells to germ cells that may promote germline differentiation is still undetermined.

## 4. Follicle Cell Patterning by Interplay of Local Signaling

Follicle cell precursors are specified into polar cells, stalk cells, and main-body follicle cells. The first step of follicle cell diversification is a Delta/Notch signaling in the germarium between germline and follicle cells [8,9]. Since a Delta/Notch pathway is a juxtacrine signaling in which a ligand and a receptor are both expressed on the plasma membrane, it absolutely requires direct cell-cell contact. 

When a germline cyst encounters follicle stem cells (FSCs) at region 2a/2b, the cyst contact induces FSC division which proliferates into polar/stalk precursors [4]. Polar/stalk precursors separate the cyst from the adjacent younger cyst in region 2b. The Notch ligand Delta expressed on the anterior germline cyst induces the adjacent anterior follicle cells to differentiate into polar cells. Mutation of Delta in the germ cells or mutation of Notch in the follicle cell precursors both result in loss of polar cells, leading to egg chamber fusion [9,42,43,44,45], indicating that the interaction of these cells via Notch/Delta signal is essential for this step. 

Polar cells secrete the JAK/STAT ligand Unpaired (Upd), which induces the follicle cells immediately anterior to the polar cells to form a stalk, which is a string of five to eight cells. Stalk cells intercalate and pinch off the egg chamber from the germarium. The younger cyst moves through the germarium and can initiate the same process again. In consequence, the neighboring egg chambers are linked by the stalk (Figure 1C) [46,47,48]. Because Unpaired is a humoral ligand, cells may secrete ligand locally to restrict the responding follicle cell population. A recent study showed that Unpaired localize to a concentration gradient on the apical surface of the follicular epithelium together with *Drosophila* HSPG, Dally [11]. 

In addition, the Hh and Wnt signaling also regulates follicle cell patterning. Over-activation of the Hh pathway delays follicle cell differentiation and causes an increase in the number of polar cells [49,50,51]. Wnt signaling transiently inhibits expression of the main body cell fate, and Wnt hyperactivity strongly biases cells towards polar and stalk fates [52]. 

In summary, multiple local signaling pathways are repeatedly turned on and off to create precisely shaped egg chamber strings. Further study is necessary to determine whether these pathways are modulated by specific cellular machinery.

## 5. Oocyte Polarization: Polarized Exocytosis of Gurken Protein

Polarized exocytosis is broadly observed in various cell types (reviewed in [53,54,55]). Using this mechanism, cells secrete protein toward a certain direction, thus preventing unfavorable interaction with non-target cells. 

A well-studied example in the *Drosophila* ovary is polarized exocytosis of Gurken, a member of the TGF-alpha family protein. Anterior/Posterior (A/P) and Dorsoventral (D/V) axes of the oocyte are specified by interactions between oocyte and follicle cells. Gurken signal first acts to establish the A/P axis then to form the D/V axis. Gurken mRNA accumulates between the oocyte nucleus and the oocyte/follicle cell interface, first near the posterior pole. After the onset of oocyte nucleus migration in stage 7–8, the Gurken mRNA re-accumulates at the dorsal-anterior region of the oocyte [56,57]. At the oocyte surface near these regions, Gurken is secreted locally and binds to Torpedo, a receptor tyrosine kinase similar to the EGF receptor, then activates EGFR signaling in nearby follicle cells (Figure 1G) [58,59]. Once follicle cells are specialized by Gurken, they signal back to reorganize the oocyte’s cytoskeleton that finally defines the embryonic axes [60]. An endoplasmic reticulum (ER) resident protein Cornichon mediates the directional transport of Gurken towards the posterior or dorsal anterior oocyte membrane [61]. In addition, Syntaxin-1A, a member of the N-ethylmaleimide-sensitive factor (NSF) attachment protein receptor (SNARE) complex, is essential for trafficking of the Gurken protein. Syntaxin-1A is associated with the Golgi membrane and regulates transportation of Gurken-containing vesicles along polarized microtubules in the oocyte and enables polarized secretion [12]. Polarized microtubules exist in networks throughout the interconnected germ cells (reviewed in [62]), which facilitate polarized secretion of several molecules including Gurken, and is regulated by the activity of the small GTPase Rab6 [12,63,64].

Polarized secretion might be also utilized in many other contexts of oogenesis discussed earlier. For example, the follicle cell epithelia and the oocyte membrane are facing each other; follicle cells may utilize a similar polarized secretion mechanism along their apical/basal polarity to send the signal to the oocyte but not to neighboring follicle cells. This signal coming from the follicle cell back to oocyte has not yet been identified.

## 6. Transfer of Vitelline Components from Follicle Cells to Oocyte: The Potential Role of Microvilli

Certain cells may transfer their product into different cells. One example is seen during vitellogenesis beginning at stage 8. The oocyte nucleus is largely transcriptionally quiescent and does not resume transcription until the zygotic genome becomes active after fertilization. Therefore, oocyte growth largely depends on stored mRNA translation and/or direct material transfer from neighboring cells. 

Maturing oocytes accumulate yolk proteins, which will serve as nutrients for the embryo. This step is called “vitellogenesis”. Yolk proteins are mainly exocytosed from follicle cells surrounding the oocyte [13], then taken up by the oocyte via clathrin-mediated endocytosis [65,66]. An early ultrastructure analysis identified microvilli projecting from the membranes of both the oocyte and the surrounding follicle cells, which interdigitate around vitelline bodies in the oocyte-follicle cell interface (Figure 1H) [2]. More recently, an immunogold EM study demonstrated that follicle cell microvilli are the site of secretion of vitelline components [67]. 

The role of oocyte microvilli during vitellogenesis is less understood. As oocyte and follicle cell microvilli interdigitate, they may make contact at various points along the oocyte-follicle cell interface. In addition to yolk production, follicle cells synthesize and secrete egg shell components: first the components of the inner layer (vitelline membrane) during stages 9–12, and then the outer layer chorion during the final stages [68,69]. Egg shell components are secreted into the space between the oocyte and follicle cells, where they are seen as large granules called vitelline bodies, which coalesce to form the vitelline membrane [2,69]. As the Yolk proteins are secreted at the same time, they must find the way to the oocyte through the oocyte-follicle cell interface which is crowded with vitelline bodies [67,70]. Interdigitating microvilli could provide a solution to this obstacle. Whether the exchange of yolk proteins occurs at the microvilli contact sites is unknown. Work has been done showing yolk proteins from the follicle cells transported through gaps in the vitelline membrane [67], although it remains unclear if these gaps represent actual microvilli or simply spaces within the vitelline membrane left by microvilli. Future studies addressing this topic could answer this question more directly using live-imaging techniques. 

Alternatively, the microvilli could be used to transduce other signaling pathways between the oocyte and follicle cells. In that case, direct contact between microvilli could serve as a site for receptor/ligand interaction. Whether any signaling molecules are interacting on microvilli remains to be determined. Similar microvillar projections have also been reported in the mouse follicle, where they are termed transzonal projections (TZPs). TZPs extend from somatic cells surrounding the oocyte and make contact with oocyte microvilli and plasma membrane, where they form gap junctions to exchange the small molecules between germline and soma [71,72,73], indicating the possibility of conservation of such mechanisms across species. 

During vitellogenesis, follicle cells secrete massive amounts of egg shell materials [2,13,69]. Early EM studies on Drosophila egg chambers identified endoplasmic reticulum (ER) in follicle cells described as “concentric lamellae or whorls of cisternal ER” [2]. These ER structures are now referred to as stacked ER sheets [74], and while we know that stacked ER sheets are present in many cells across species-especially in secretory cells-the mechanism by which they form is completely unknown. It is hypothesized that stacked ER sheets are advantageous to secretory cells by providing the large ER membrane surface area within a limited volume of cytoplasm which permits maximal synthesis of secreted proteins [74]. 

Stacked ER sheets are seen in close association with precursors to vitelline bodies-the granules that collect in the oocyte-follicle interface and coalesce to form the vitelline membrane. Stacked ER sheets are present in follicle cells only during stages 9–12 [2], when vitellogenesis takes place and follicle cells secrete vitelline material. This suggests that stack formation can be developmentally regulated and assembled or disassembled to meet the needs of the cell. Very little is known about the mechanisms of morphogenic reorganization and the functional difference among distinct ER morphologies. As we can follow the stage-specific reorganization of ER structure, *Drosophila* follicle cells may serve as one of the best in vivo models to answer these questions.

## 7. Nurse Cell Dumping

The last extreme example of cell-cell interaction observed in *Drosophila* oogenesis is “nurse cell dumping,” which occurs during stage 11 when nurse cells rapidly transport their cytoplasm into the oocyte. Actin and microtubules within the cytoskeleton are both required for dumping machinery [20]. Before this rapid dumping occurs, cytoplasm flows from the nurse cells to the oocyte through cytoplasmic bridges called ring canals which connect nurse cells and the oocyte [75,76,77]. Ring canals are circular structures composed of several proteins including actin and actin-binding proteins [19,78,79,80], and the adducin-like *hts* protein [77], which are required both for ring canal formation and for cytoplasmic flow from the nurse cells to the oocyte [17,77,79,80]. Nurse cells produce specific cargos such as mRNA and organelles which are required for proper function of the oocyte and embryo patterning [16], and are transported to the oocyte through ring canals along microtubule tracks (Figure 1D) [81,82,83]. During dumping at stage 11, all remaining nurse cell cytoplasm is rapidly transferred to the oocyte within only around 30 minutes [68,76]. This process is facilitated by actin-myosin contraction [84]. At stage 10B, just preceding dumping, the nurse cell actin network undergoes extensive remodeling, an event triggered by prostaglandin signaling by the COX-like enzyme Pxt [85,86,87]. Bundles of parallel actin filaments extend from the intracellular plasma membrane of nurse cells to their nuclei, in association with perinuclear actin [88,89]. During dumping, as the nurse cells shrink while donating their cytoplasm, these actin bundles physically prevent the nuclei from blocking the intercellular canals, thus allowing complete transfer of nurse cell cytoplasm to the oocyte.

Concurrent with dumping, nurse cells degenerate at the anterior tip of the egg chamber via programmed cell death which requires the caspase Dcp-1 [90]. Dcp-1 mutant nurse cells not only fail to undergo programmed cell death, but also exhibit a “dumpless” phenotype, failing to rapidly transport their cytoplasm to the oocyte during stage 11, suggesting a link between the processes of programmed cell death and nurse cell dumping. However, the canonical apoptotic genes *reaper*, *hid*, and *grim* are not required for the programmed death of nurse cells [91], suggesting that an alternative pathway is involved. Recent work has shown that the somatic follicle cells surrounding the nurse cells (called stretch follicle cells) are required for nurse cell dumping and death, via the c-Jun N-terminal kinase (JNK) signaling pathway [92,93]. The JNK pathway is active in stretch follicle cells during stage 11 [94], and is upregulated during stages 12–13, where it is required specifically in follicle cells for the death of nurse cells [93]. Nurse cell death, however, is nonsynchronous, and it is unclear exactly how somatic stretch follicle cells facilitate the apparent phagoptosis of germline nurse cells. Furthermore, the way in which nurse cell dumping depends on nurse cell death, and the interplay of germline and soma in regulating these processes, remains unknown. 

## 8. Conclusions, Implications, and Future Directions

Here we briefly summarized local cell-cell interactions occurring during *Drosophila* oogenesis, compiling both recent and old studies on the specialized cellular machineries and structures which are indispensable for signaling. Modification of signaling by such mechanisms may be critical for signal specificity, efficiency and adaptability. Membrane protrusions, including cytonemes, cytosensors, filopodia and microvilli are still poorly characterized structures. However, recent imaging techniques have started identifying more and more signaling protrusions in broad tissue contexts across species. Interestingly, cytonemes possess a feedback mechanism to flexibly respond the tissue requirement, suggesting the intriguing possibility that signaling protrusions may be the key structure that can buffer challenging physiological conditions. Further study will be necessary to test this hypothesis and to determine whether it represents a conserved mechanism for protrusion-mediated signaling.

## Figures and Tables

**Figure 1 cells-09-00274-f001:**
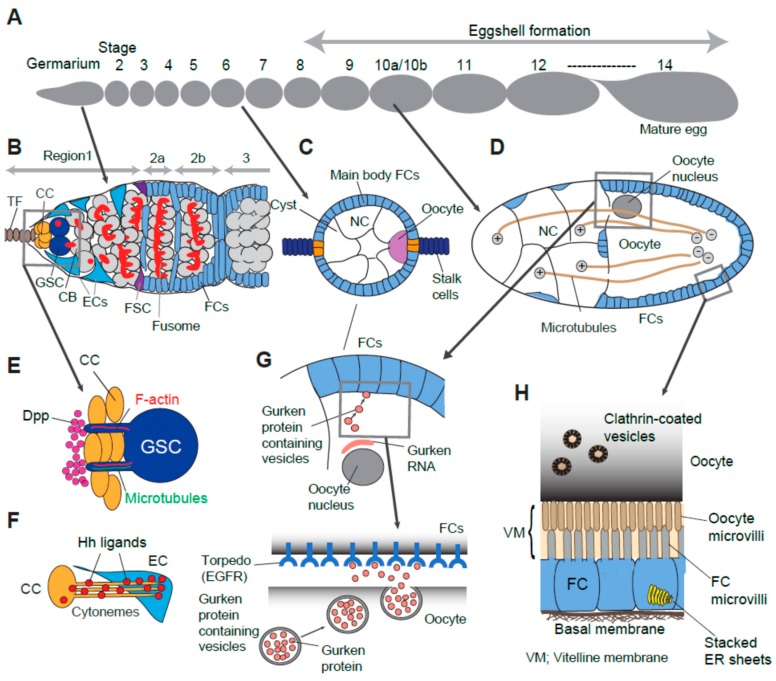
Examples of usage of specialized structures and/or machineries for cell-cell interaction. (**A**) Overview of fourteen developmental stages of oogenesis. (**B**) Cellular organization of germarium. (**C**) Stage 6 egg chamber. Cyst contains 15 nurse cells and one posteriorly localized oocyte. Specified follicle cells (Polar, stalk and main body follicle cells) are determined by interplay of local signaling (See Section 4). (**D**) Stage 10b egg chamber. Nurse cells directly transfer their material (mRNA, organelle, protein) toward oocyte via microtubule tracks. Microtubule plus end are in nurse cells and their minus ends are in the oocyte. (**E**) A GSC with a cytocensor and an actin protrusion. Dpp ligands accumulate at the anterior face of CCs away from GSCs. (**F**) CC emanates cytonemes to deliver the Hh ligand toward EC. (**G**) Polarized Gurken secretion in a stage 10b egg chamber. The oocyte nucleus is positioned at the dorsal-anterior region of oocyte. Gurken mRNA are seen between the oocyte nucleus and oocyte-follicle cell interface. Polarized secretion of Gurken occurs locally at the dorso-anterior surface of oocyte and activates the Torpedo receptor expressed on the surface of nearby follicle cells. (**H**) Yolk material deposition from follicle cell into the oocyte (See details in Section 6). Oocyte microvilli and follicle cell microvilli extend and interdigitate each other between the space of the oocyte and follicle cell layer which is filled with vitelline membrane. CC (cap cells), TF (terminal filament), GSC (germline stem cell), CB (cystoblast), EC (escort cell), FSC (follicle stem cell), FCs (follicle cells), NC (nurse cell), Dpp (decapentaplegic), Hh (hedgehog), EGFR (epidermal growth factor receptor), VM (vitelline membrane).

**Figure 2 cells-09-00274-f002:**
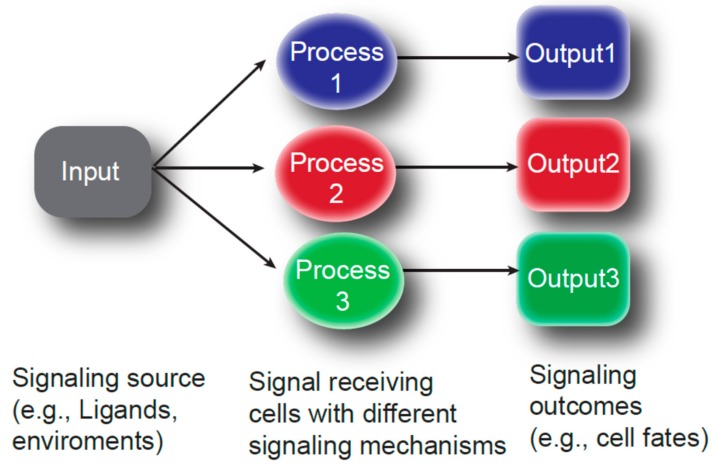
Different cellular processes modulate signaling outcome. Only a handful of signaling pathways (input) are known to regulate multiple processes of oogenesis. Therefore, choosing a specific receptor-ligand combination may not be the only solution for specificity of the signaling. Modification of the signal (process) may determine the population of responding cells and may modulate cellular responses (output).

**Table 1 cells-09-00274-t001:** Modification of cell-cell interaction by cellular/extracellular components.

Sending Cell	Receiving Cell	Signaling Pathway	Cellular/Extracellular Components		Function	References
Cap cells	GSCs	BMP	Filopodia Cytocensors	Signal reception and attenuation		[6]
			HSPG (Dally)	Local Dpp tethering		
Cap cells	Escort cells	Hh	Cytonemes	Hh delivery		[7]
Anterior Cyst	Follicle cells ->Polar cells	Delta/Notch	Contact dependent signaling	Local polar cell specification		[8,9]
Escort cells	Cystoblasts, Cyst	EGFR JAK/STAT	Microtubule rich membrane extension	Posterior allocation of the cyst (?)		[10]
Polar Cells	Follicle cells -> Stalk cells	JAK/STAT	HSPG (Dally)	Local Unpaired gradient		[11]
Oocyte	Dorsoventral Follicle cells	Gurken/Torpedo (EGFR)	Polarized Gurken secretion (Exocyst)	Signal localization		[12]
Follicle cells	Oocyte	N/A	Exo/endo cytosis Microvilli	Vitellogenesis		[13,14,15]
Nurse cells	Oocyte	N/A	Ring canal Microtubules	Cytoplasm transfer		[16,17,18]
			Actin/Myosin	Dumping		[19,20,21]

BMP (bone morphogenetic protein), HSPG (heparan sulfate proteoglycan), Hh (hedgehog), EGFR (epidermal growth factor receptor), JAK/STAT (Janus kinase/signal transducer and activator of transcription), N/A (not applicable).

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
