# Peer review of "Modulation of Cell–Cell Interactions in Drosophila Oocyte Development"

_cells, 2020, doi:10.3390/cells9020274_

Round 1
Reviewer 1 Report
In this comprehensive mini review, the authors provide a cumulative update regarding cell-cell interactions during Drosophila oogenesis. The manuscript manages to concisely present the current knowledge, focusing mainly on early oogenesis.
Please find some comments/suggestions as per below, in order to improve the quality of the article:
My main concern is that the manuscript, in several occasions, is rather incomprehensive, due to either very short (declaration-like) or elaborated sentences.
The authors should also pay a little more attention to details. For example, in line 29, it is implied that all 14 egg chamber stages are simultaneously observed in a single ovariole, which is not the case (i.e https://doi.org/10.1046/j.1365-2435.1998.00224.x and https://doi.org/10.1038/srep18850).
In Fig. 1A, in which the stages of egg chamber development are depicted, stages 8-14 are all described as vitellogenic stages. Nevertheless, since the vitelline membrane, the first eggshell layer, is synthesized during stages 8–10 of oogenesis (https://doi.org/10.1016/S0074-7696(00)98003-3 , https://doi.org/10.1007/bf00324615 the word vitellogenesis should rather be replaced by a phrase such as eggshell formation/production/morphogenesis.
Regarding the microtubule polarity in Fig. 1D (depicting a stage 10B egg chamber) the authors should provide a reference supporting the shown polarity. Microtubule polarity in the Drosophila oocyte is described in the following review article (https://doi.org/10.1002/dvdy.20770).
The differentiation, after stage 7, of epithelial follicle cells into the five main epithelial fates: border, stretched, centripetal, posterior, and main body follicle cells (https://doi.org/10.1002/dvdy.21625) should be discussed in the text before nurse cell dumping section, in which the stretch cells term arises.
Finally, cell interactions that occur during late oogenesis, such as border cell migration and micropyle formation, dorsal appendages morphogenesis and egg chamber elongation are not discussed in this Review.
Minor comments
The citations in some cases are annotated at the end of each paragraph and not at the end of the relevant sentence, making it hard for the reader to find the appropriate reference. It’s generally recommended to allocate citations to the specific sentences.
In the second paragraph of the review (lines 29-37), the presented text is not supported by any citation.
The Table, at least in my PDF version, is not well aligned. The addition of horizontal lines may improve the presentation.
Please check if all abbreviated text in Fig.1 is explained in its legend (i.e. FSC).
Line 20: in vivo should be italics.
The syntax of the sentence in lines 134-136 is not correct and needs editing.
Line 173: the “an” before oocyte in my opinion should be replaced by “the”.
Author Response
Point 1: My main concern is that the manuscript, in several occasions, is rather incomprehensive, due to either very short (declaration-like) or elaborated sentences.
Response: We thank the reviewer for their comment. We have expanded on points especially raised by reviewers and added references.
Point 2: The authors should also pay a little more attention to details. For example, in line 29, it is implied that all 14 egg chamber stages are simultaneously observed in a single ovariole, which is not the case (i.e https://doi.org/10.1046/j.1365-2435.1998.00224.x and https://doi.org/10.1038/srep18850).
Response: We have edited the description of egg chambers to be more accurate.
Point 3: In Fig. 1A, in which the stages of egg chamber development are depicted, stages 8-14 are all described as vitellogenic stages. Nevertheless, since the vitelline membrane, the first eggshell layer, is synthesized during stages 8–10 of oogenesis (https://doi.org/10.1016/S0074-7696(00)98003-3 , https://doi.org/10.1007/bf00324615 the word vitellogenesis should rather be replaced by a phrase such as eggshell formation/production/morphogenesis.
Response: The figure has been edited accordingly.
Point 4: Regarding the microtubule polarity in Fig. 1D (depicting a stage 10B egg chamber) the authors should provide a reference supporting the shown polarity. Microtubule polarity in the Drosophila oocyte is described in the following review article (https://doi.org/10.1002/dvdy.20770).
Response: We thank the reviewer for their suggestion. We have added a point on microtubule polarization and included the reference.
Point 5: The differentiation, after stage 7, of epithelial follicle cells into the five main epithelial fates: border, stretched, centripetal, posterior, and main body follicle cells (https://doi.org/10.1002/dvdy.21625) should be discussed in the text before nurse cell dumping section, in which the stretch cells term arises.
Response: Thanks for pointing this out. We have added a description of stretch follicle cells before they are mentioned in nurse cell dumping.
Point 6: Finally, cell interactions that occur during late oogenesis, such as border cell migration and micropyle formation, dorsal appendages morphogenesis and egg chamber elongation are not discussed in this Review.
Response: We thank the reviewer for this comment. While we agree that these cell interactions in these stages do involve in the oogenesis, such processes are extensively studied as the model for epithelial reorganization and regulation and have been reviewed extensively elsewhere. Thus, we decided not to include these as we wanted to focus our review on interactions concerning the oogenesis specific processes.
Point 7: The citations in some cases are annotated at the end of each paragraph and not at the end of the relevant sentence, making it hard for the reader to find the appropriate reference. It’s generally recommended to allocate citations to the specific sentences.
Response: Thank you for this suggestion. We have rearranged some citations to more accurately reflect the statements made.
Point 8: In the second paragraph of the review (lines 29-37), the presented text is not supported by any citation.
Response: We have added citations.
Point 9: The Table, at least in my PDF version, is not well aligned. The addition of horizontal lines may improve the presentation.
Response: We have added lines in the table.
Point 10: Please check if all abbreviated text in Fig.1 is explained in its legend (i.e. FSC).
Response: We have defined abbreviations in the legend.
Point 11: Line 20: in vivo should be italics.
Response: We revised it.
Point 12: The syntax of the sentence in lines 134-136 is not correct and needs editing.
Response: We have edited the sentence.
Point 13: Line 173: the “an” before oocyte in my opinion should be replaced by “the”.
Response: We have made the edit.
Reviewer 2 Report
The review does a good job covering the cell-cell signaling events regulating processes throughout Drosophila oogenesis. However, the discussion of late events in oogenesis can be significantly improved (see major issues below). Additional minor comments/concerns are listed below.
Major issues:
The section on nurse cell dumping does a poor job describing the actin structures necessary for the rapid transport of the nurse cell contents into the oocyte. Additionally, these actin remodeling events are regulated by prostaglandin signaling. It seems like an oversite to not discuss the signaling and cell-cell interactions mediating ovulation.
Minor issues:
There are numerous grammatical errors throughout. “All 14 stages of Drosophila oogenesis can be observed in a single ovariole (Figure 1A).” This is not true. Ovarioles only have about 8 follicles in them. “In region1, located at the 3 germarium tip, 2-3 germline stem cells (GSCs) contact a cluster of somatic cap cells. A GSC divides asymmetrically to self-renew and give rise to a cystoblast. The cystoblast undergoes four rounds of 34 incomplete divisions to generate an interconnected 16-cell germline “cyst”.” From the text it is unclear where the 16 cell cysts are in the germaruim. “Each follicle cell type signals back to the germline cyst to support the oocyte growth, maturation, and polarization.” This statement is vague and unclear, in particular because you are talking about cysts but referring to figures on follicles. Also the figure reference should be Fig. 1G-H. Stage should be capitalized when stating Stage 9, etc. Line 93 – it is unclear that there are two protrusions, one actin-based and one microtubule rich. Line 95 – what does “simply to reach out to Dpp” mean? Clarity in the text is needed. Line 106 – what do you mean by “anterior-most escort cells couple with GSCs” Line 112 – did you mean anterior escort cells and not apically located? Lines 131-132 – “explaining how escort cells maintain extensive contacts with germ cells, while allowing a posterior allocation of differentiating GSC-descendants.” This statement is unclear – is the first part talking about GSCs? Posterior allocation is very odd phrasing and is not clear. Lines 187-188- “ follicle cells may utilize similar polarized secretion mechanisms along their apical/basal polarity.” to do what? Line 191 – vitellogenesis is not solely in late oogenesis – it starts at Stage 8. I suggest being specific about the stages being discussed. The “Stacked ER…” section seems like a tangent and could be shortened and combined with the vitellogenesis section.
Author Response
Point 1: The section on nurse cell dumping does a poor job describing the actin structures necessary for the rapid transport of the nurse cell contents into the oocyte. Additionally, these actin remodeling events are regulated by prostaglandin signaling.
Response: We thank the reviewer for their comment. We have added a description of actin structures during dumping and references.
Point 2: It seems like an oversite to not discuss the signaling and cell-cell interactions mediating ovulation.
Response: We thank the reviewer for this suggestion. We are aware of several important works have been done for cell-cell interaction mediating ovulation. However, in this review, we aimed to focus on the processes for the oocyte development and therefore did not include the steps of ovulation. We have edited title and abstract to more accurately represent our scope.
Point 3: “All 14 stages of Drosophila oogenesis can be observed in a single ovariole (Figure 1A).” This is not true. Ovarioles only have about 8 follicles in them.
Response: We have corrected this error.
Point 4: “In region1, located at the 3 germarium tip, 2-3 germline stem cells (GSCs) contact a cluster of somatic cap cells. A GSC divides asymmetrically to self-renew and give rise to a cystoblast. The cystoblast undergoes four rounds of 34 incomplete divisions to generate an interconnected 16-cell germline “cyst”.” From the text it is unclear where the 16 cell cysts are in the germarium.
Response: We have edited the sentence to reflect this.
Point 5: “Each follicle cell type signals back to the germline cyst to support the oocyte growth, maturation, and polarization.” This statement is vague and unclear, in particular because you are talking about cysts but referring to figures on follicles. Also the figure reference should be Fig. 1G-H.
Response: We have edited this sentence and updated the figure reference for clarity.
Point 6: Line 106 – what do you mean by “anterior-most escort cells couple with GSCs” Line 112 – did you mean anterior escort cells and not apically located?
Response: We thank the reviewer for pointing this error out and have edited apical to anterior.
Point 7: Lines 131-132 – “explaining how escort cells maintain extensive contacts with germ cells, while allowing a posterior allocation of differentiating GSC-descendants.” This statement is unclear – is the first part talking about GSCs? Posterior allocation is very odd phrasing and is not clear.
Response: We have edited this paragraph for clarity.
Point 8: Lines 187-188- “ follicle cells may utilize similar polarized secretion mechanisms along their apical/basal polarity.” to do what?
Response: We have added to this description for clarity.
Point 9: Line 191 – vitellogenesis is not solely in late oogenesis – it starts at Stage 8. I suggest being specific about the stages being discussed.
Response: We thank the reviewer for the suggestion and have edited this part and several other throughout the review to be more specific about stages.
Point 10: The “Stacked ER…” section seems like a tangent and could be shortened and combined with the vitellogenesis section.
Response: We agree with the reviewer and have edited and combined the section with the discussion on vitellogenesis.
Reviewer 3 Report
I enjoyed reading this manuscript. It has a refreshing angle and summarises interesting work that has been recently done on the Drosophila oocyte system with a focus on cell-cell signalling.
The text flows well, there are a few minor points that I listed below. I am not sure if the format of the journal allows that, but one or two schematics could be added that help emphasise the angle the authors are trying to make.
Line 30: “apical” change to anterior
Line 31:”undifferentiated germ cells” this refers to stem cells?
Line 46:” In other words, different combinations of the cells within the same tissue sometimes use the same signaling pathway, indicating that spatiotemporal modification of signaling is critical to prevent communication between the wrong partners.” This is an interesting idea. Could the signalling pathways that the authors have in mind, be depicted in a diagram? That would be helpful
Line 134” long term live imaging?
Line 136: “Whether such dynamic membrane reorganization also regulates receptor mediated signaling between germline and soma remains unknown.” This signalling idea lacks introduction
Line 173: between the oocyte nucleus?
Paragraph 4: This is traditionally one of the first cell cell signaling events reported in that system, may be that could be highlighted? The “backsignall” is still elusive but there is a little bit more information available. Cite recent literature here or a review that details that. Rab6 also plays in Grk signalling
Author Response
Point 1: Line 30: “apical” change to anterior
Response: We thank the reviewer for pointing out this error and have corrected it .
Point 2: Line 31:”undifferentiated germ cells” this refers to stem cells?
Response: We have edited the sentence for clarity.
Point 3: Line 46:” In other words, different combinations of the cells within the same tissue sometimes use the same signaling pathway, indicating that spatiotemporal modification of signaling is critical to prevent communication between the wrong partners.” This is an interesting idea. Could the signalling pathways that the authors have in mind, be depicted in a diagram? That would be helpful
Response: We thank the reviewer for the comment and have added a generalized diagram to reflect this.
Point 3: Line 134” long term live imaging?
Response: We thank the reviewer for pointing out this mistake and have corrected it.
Point 4: Line 136: “Whether such dynamic membrane reorganization also regulates receptor mediated signaling between germline and soma remains unknown.” This signalling idea lacks introduction
Response: We have edited and rearranged this text for clarity.
Point 5: Line 173: between the oocyte nucleus?
Response: We thank the reviewer for pointing out this error and have corrected it.
Point 6: Paragraph 4: This is traditionally one of the first cell cell signaling events reported in that system, may be that could be highlighted? The “backsignall” is still elusive but there is a little bit more information available. Cite recent literature here or a review that details that. Rab6 also plays in Grk signalling
Response: We thank the reviewer for their comments and have added references and the description of Rab6 function.